# How Anesthetic, Analgesic and Other Non-Surgical Techniques During Cancer Surgery Might Affect Postoperative Oncologic Outcomes: A Summary of Current State of Evidence

**DOI:** 10.3390/cancers11050592

**Published:** 2019-04-28

**Authors:** Patrice Forget, Jose A. Aguirre, Ivanka Bencic, Alain Borgeat, Allessandro Cama, Claire Condron, Christina Eintrei, Pilar Eroles, Anil Gupta, Tim G. Hales, Daniela Ionescu, Mark Johnson, Pawel Kabata, Iva Kirac, Daqing Ma, Zhirajr Mokini, Jose Luis Guerrero Orriach, Michael Retsky, Sergio Sandrucci, Wiebke Siekmann, Ljilja Štefančić, Gina Votta-Vellis, Cara Connolly, Donal Buggy

**Affiliations:** 1Anesthesiology and Perioperative Medicine, Universitair Ziekenhuis Brussel, Vrije Universiteit Brussel, Laarbeeklaan 101, 1090 Brussels, Belgium; 2Anesthesiology, Balgrist University Hospital Zurich, 8091 Zurich, Switzerland; jose.aguirre@balgrist.ch (J.A.A.); alain.borgeat@balgrist.ch (A.B.); 3University Hospital for Tumors, Sestre Milosrdnice University Hospital Center, Zagreb 10000, Croatia; ivanka.bencic@gmail.com; 4Department of Pharmacy, Unit of General Pathology, Center on Aging Sciences and Translational Medicine (CeSI-MeT), “G. d’Annunzio” University of Chieti-Pescara, 66100 Chieti, Italy; alessandro.cama@unich.it; 5Department of Surgery, Royal College of Surgeons in Ireland, Beaumont Hospital, 9 Dublin, Ireland; ccondron@rcsi.ie; 6Department of Anesthesiology and Intensive Care, University of Linköping, 581 83 Linköping, Sweden; christina.eintrei@liu.se; 7INCLIVA Biomedical Research Institute, 46010 Valencia, Spain; pilar.eroles@uv.es; 8Biomedical Research, Network in Breast Cancer (CIBERONC), Instituto de Salud Carlos III, 28029 Madrid, Spain; 9Physiology and Pharmacology, Karolinska Institutet, Perioperative Medicine and Intensive Care, Karolinska Hospital, 171 76 Stockholm, Sweden; Anil.gupta@sll.se; 10Division of Systems Medicine, School of Medicine, University of Dundee, Dundee DD1 9SY, UK; t.g.hales@dundee.ac.uk; 11Head Department of Anesthesia and Intensive Care, Iuliu Hatieganu University of Medicine and Pharmacy, Cluj-Napoca, Romania, Outcome Research Consortium, Cleveland, OH 44195, USA; dionescuati@yahoo.com or daniela_ionescu@umfcluj.ro; 12Department of Anesthesia, Fiona Stanley Hospital, Perth, Western Australia. University College Dublin School of Medicine and Medical Science, 4 Dublin, Ireland; markzjohnson@gmail.com; 13Department of Surgical Oncology, Medical University of Gdańsk, 80-210 Gdańsk, Poland; pawel.kabata@gmail.com; 14Surgical Oncology, University Hospital for Tumors, Sestre Milosrdnice University Hospital Center, Zagreb 10000, Croatia; iva.kirac@kbcsm.hr; 15Anesthetics, Pain Medicine & Intensive Care, Department of Surgery and Cancer, Imperial College London, Chelsea & Westminster Hospital, London SW10 9NH, UK; d.ma@imperial.ac.uk; 16San Gerardo University Hospital, Monza, Italy. Clinique Saint Francois, 36000 Chateauroux, France; zheri@hotmail.com; 17Institute of Biomedical Research in Malaga [IBIMA], Department of Cardio-Anaesthesiology, Virgen de la Victoria University Hospital, 2010 Malaga, Spain; guerreroorriach@gmail.com; 18Department of Pharmacology and Pediatrics, School of Medicine, University of Malaga, 29071 Malaga, Spain; 19Department of Environmental Health, Harvard TH Chan School of Public Health, Boston, MA 02115, USA; michael.retsky@gmail.com; 20Visceral Sarcoma Unit, CDSS—University of Turin, 10124 Turin, Italy; sergio.sandrucci@unito.it; 21Department of Anesthesiology and Intensive Care, Örebro University, 702 81 Örebro, Sweden; wiebke.siekmann@regionorebrolan.se; 22Intensive Care Unit, University Hospital for Tumors, Sestre Milosrdnice University Hospital Center, Zagreb 10000, Croatia; lstefancic58@gmail.com; 23Departments of Anesthesiology and Surgery, College of Medicine, University of Illinois at Chicago, Chicago, IL 60607, USA; ginavot@uic.edu; 24Mater Misericordiae University Hospital, Eccles st., D07 R2WY Dublin, Ireland; caraconnolly83@gmail.com; 25Mater University Hospital, School of Medicine, University College Dublin, 4 Dublin, Ireland; donal.buggy@ucd.ie; 26Anaesthesiology & Perioperative Medicine, Mater University Hospital, School of Medicine, University College Dublin, Ireland and Outcomes Research Consortium, Cleveland Clinic, OH 44195, USA

**Keywords:** cancer, anesthesia, analgesia

## Abstract

The question of whether anesthetic, analgesic or other perioperative intervention during cancer resection surgery might influence long-term oncologic outcomes has generated much attention over the past 13 years. A wealth of experimental and observational clinical data have been published, but the results of prospective, randomized clinical trials are awaited. The European Union supports a pan-European network of researchers, clinicians and industry partners engaged in this question (COST Action 15204: Euro-Periscope). In this narrative review, members of the Euro-Periscope network briefly summarize the current state of evidence pertaining to the potential effects of the most commonly deployed anesthetic and analgesic techniques and other non-surgical interventions during cancer resection surgery on tumor recurrence or metastasis.

## 1. Introduction

Surgery (under anesthesia) is an important part of modern health care, helping millions of people lead healthier, more productive lives [1]. More than 310 million major surgical procedures are performed worldwide every year [2]. While the value of surgery is clear, little is known about the impact of anesthesia or other perioperative factors on subsequent health status and quality of life, which can sometimes be compromised. For example, surgical tumor excision is the primary approach to breast, colon, brain, prostate, and several other cancers [3]. 

Preclinical and clinical studies support the hypothesis that anesthesia influences cancer biology and outcomes. Although there is plenty of retrospective published data and some prospective randomized studies available and ongoing, these are not sufficiently conclusive to suggest changes in clinical practice. The strongest support for a role of anesthetics on cancer biology comes from preclinical studies. However, the literature is sparse, often conflicting, and preclinical studies cannot fully reproduce the complexity of human physiology during the perioperative period. 

A previous consensus statement was published in 2014 but has not been updated [4]. In 2016, the EU supported a COST Action (Euro-Periscope, 15204), a network of researchers, industry partners, and clinicians interested in this question, to form a pan-European collaboration. In the current narrative review, members of Euro-Periscope briefly summarize the current state of evidence pertaining to the potential effects of the most commonly deployed anesthetic and analgesic techniques and other perioperative non-surgical interventions during cancer resection surgery on tumor recurrence or metastasis. After two-rounds of consultation with all of the members, aiming to provide a more complete vision, updates were proposed and consensually accepted by the entire panel. The highest level of evidence available illustrates each topic.

## 2. Update on Methodology

### Anesthesia and Cancer Research: Need for Translational Prospective Studies and Methodological Adaptations

Globally, research into the impact of anesthesia on cancer during the last twenty years has focused either at the pre-clinical or clinical level. This disconnected approach has led to inconsistent selection of endpoints and a limited perspective in a poorly understood field. A similar problem has been highlighted in perioperative pain research [5]. Recently, a more consistent approach towards defining oncological endpoints was published [6,7].

Inevitably, in some cases, the quality of biological studies could be improved by ensuring an adequate number of experiments to provide more robust statistical analysis. Likewise, for translational studies ideally testing a single primary hypothesis, the sample size should be calculated according to the true variance of the data and the minimum biological effect.

Transcriptomic and genomic studies testing multiple biological hypotheses cannot be interpreted only from *p*-values. Therefore, it is recommended that sample size should be calculated and data analysis performed according to the tolerable False Discovery Rate (FDR = proportion of false positives among the declared significant results) and False Negative Rate (FNR, or sensitivity) [8]. As FDR depends on sample size, it is proposed by some authors to include a minimum of 50 subjects per group [8].

Ideally, translational projects should combine omics approaches in conjunction with epidemiological confirmation of the disease as well as clinical trials, when appropriate. Preferably, this should be done in combination with analyses of clinically relevant endpoints and biologically promising biomarkers. This may permit identification of specific mechanisms that could be addressed, such as perioperative angiogenesis and neuroimmune interactions. Last but not least, these interventions may take place in multi-arm/platform trials, particularly in complex perioperative settings.

## 3. Update on Biology

### 3.1. Perioperative Angiogenesis in Breast Cancer

Catecholamines, prostaglandins, and angiogenic factors are secreted abundantly during the perioperative period in response to stress and surgery and were shown by translational studies as possible promoters of tumor metastasis. Circulating tumor cells are seen in all cancers and transient systemic inflammation accompanying surgery can enhance capillary leakage facilitating angiogenesis of dormant micrometastasis, the proliferation of dormant cells and seeding of circulating cancer stem cells resulting in early relapse.

Epidemiologic studies suggest that the choice of anesthetics may affect recurrence in breast cancer through changes in angiogenesis. Studies in a xenograft model showed that antiangiogenic therapy in the perioperative or direct postoperative period following primary breast tumor resection reduces progression of bone metastasis [9].

Perioperative administration of the NSAID ketorolac (non-steroidal anti-inflammatory drug) has been associated with a significant reduction in early recurrence following breast cancer surgery [10]. NSAID may interfere with the SDF1-CXCR4 axis in the cyclo-oxygenase type 2-prostaglandins pathway (COX2-PGE-SDF1).

Comparing general anesthesia with sevoflurane-opioids versus propofol-paravertebral block showed that the latter inhibits cancer cell proliferation, reduces the stress response, as well as metalloproteinases and transforming growth factor β [11]. Opioids may be proangiogenic through an immunosuppressive effect and can also transactivate VEGF receptor by Src activation [12]. Similarly, sevoflurane may promote angiogenesis by increasing VEGF-C serum concentrations, whereas propofol can have an anti-angiogenic effect [12]. 

However, using analgesic doses of morphine and other opioids in preclinical studies in mouse models for HER2+ breast cancer found no adverse effect on tumor metastatic dissemination, angiogenesis and the concentration of tumor-infiltrating immune cells after surgery, suggesting that opioid analgesics can be used safely [13].

In summary, there are no conclusive studies relating to angiogenesis in breast cancer patients that should change our clinical practice. 

### 3.2. Anesthetic Neuromodulation and Cancer Biology

The nervous system is a primary actor in early and late cancer development. Many cancers e.g., colon and prostate present a specific growth pattern called “perineural invasion” which is driven by neurons. The growth of some cancers like prostate, pancreas, and breast cancer, is associated with “neurogenesis” and an abundant infiltration of the tumor by the autonomic nervous system. The stress response, results from stimulation of the sympathetic system, which enhances tumor growth through immunosuppression and beta-adrenergic receptor stimulation, expressed on tumor cells [14]. In animal studies, sympathectomy slows cancer progression, and human retrospective data suggest that cancer patients taking beta-blockers have lower recurrence rates and lower mortality [15]. The parasympathetic system modulates cancer progression through cholinergic receptors and the vagus nerve. Stimulation of the vagus nerve slows tumor progression by reducing sympathetic activity and oxidative stress, and by modulating the innate and adaptive immune responses [16]. Finally, sensory neurons and the tumor micro-environment are strongly involved in the modulation of inflammation, in local tissue homeostasis, in healing and development, especially under conditions of psychological stress. The activation of sensory neurons during pain enhances tumor progression and metastatic potential. Regional anesthesia blocks somatic nociception and inhibits sympathetic preganglionic outflow (functional sympathectomy) during surgery. Moreover regional anesthesia, by blocking sympathetic nervous system output, induces a prevalence of parasympathetic tone. Local anesthetics can also modulate autonomic receptors [17]. For these reasons, more studies are needed to investigate the action of regional anesthetic neuromodulation on cancer progression.

### 3.3. Surgery and Immunomodulation

Surgery evokes an acute-phase response, described above as the stress response that includes a transient immunosuppression and alterations in gastrointestinal function and there is clear evidence that sarcopenia, malnutrition, arginine bioavailability, and a high arginase activity state contributes to adverse surgical outcomes. Approaches to improve this will be discussed later. In short, approaches include pharmaconutrition, aimed at improving immune function (termed immunonutrition) and both parenteral and enteral approaches including amino acids (arginine and glutamine), lipids (omega-3 fatty acids), micronutrients (vitamins C and E), and nucleotides [18]. The benefits of perioperative treatment with immunonutrients are well documented and include reduction of postoperative complications, reduced loss of skeletal mass, modulated inflammatory response, and improved protein synthesis after surgery [19,20,21,22,23,24,25,26,27,28,29,30,31,32]. It is known that taurine supplementation augments immune function and protects the gut from intestinal barrier dysfunction induced by surgery [33]. The same phenomenon of complex immune dysfunction is seen in cancer patients and clinical data show a correlation between high-density leukocytic infiltration in solid tumors and poor patient outcomes. Inflammatory cells produce pro-angiogenic mediators and extracellular proteases that influence metastases by inducing tissue remodeling, and tumor angiogenesis [34]. Resolvin D1, a metabolite of Omega-3 fatty acids through the Cox-2 pathway has been shown to enhance endogenous clearance of tumor cell debris following chemotherapy thus complementing cytotoxic cancer therapies. Perioperative treatment with Resolvin D1 in a murine model reduces lung metastases upon surgical excision of primary breast tumors in a murine model [35]. However, the role for immunomodulating formulas within the enhanced recovery pathway for cancer patients undergoing surgery remains unclear. 

### 3.4. Animal Studies of Perioperative Drugs in the Setting of Cancer

As the scientific investigation of the potential effects of perioperative agents on cancer evolves from retrospective analysis of human trials to potential prospective work, animal investigations can play an important role in refining potential therapeutic strategies for such studies. Animal models of tumors have numerous advantages and drawbacks when compared with human research. Animal models frequently have better control of variables and deliver faster results than human studies. There is, however, no perfect non-human cancer model; they may lack many of the biological steps of tumor development and metastasis as well as not necessarily reflecting the complex immune system and tumor interplay seen in human disease. Although there have been limited animal studies to date, lidocaine infusion has been implicated in having a beneficial role in the perioperative setting and as a chemotherapeutic agent in the non-operative setting [36]. Propofol has been shown to be protective or non-inferior to controls using in vivo models [36]. Some perioperative agents have shown signals that may be potentially harmful, or promote metastasis, in the setting of cancer. Such agents include corticosteroids and dexmedetomidine, which have been shown to increase metastasis when studied in laboratory experiments in the setting of malignancy [37] but not in other in vivo experiments and large retrospective series [38,39,40,41]. Results of the translational work should then be interpreted with caution due to the small numbers of studies and the intrinsic weaknesses associated with in vivo models. However, they are useful in hypothesis development and potential strategies for improving outcomes in human disease. 

## 4. Update on the Available Evidence Regarding Anesthetic/Analgesic Technique

### 4.1. Metastasis: A Role for Amide-Type Local Anesthetics?

Metastases account for the great majority of cancer-associated deaths, yet this complex process remains the least understood aspect of cancer biology. The dissemination of cancer cells from primary tumors and subsequent seeding of new tumor colonies in distant tissues involves a multistep process known as the invasion-metastasis cascade, in which the Src system seems to play an important role [42]. Lidocaine at clinically achievable concentrations has been shown in vitro to statistically inhibit the phosphorylation of Src and the ICAM-1 expression in human lung cancer cells [43]. This study also demonstrated that the effect was independent of the function of the sodium ion channel. Moreover, lidocaine has also been shown to inhibit NaV1.5 sodium ion channels [44], and the TRPV6 receptors [45], inhibit the activity of epidermal growth factor in humans with tongue cancer [46], induce apoptosis and suppress tumor growth in human hepatocellular cells in vitro [36], and cause time- and dose-dependent demethylation of deoxyribonucleic acid in breast cancer cell lines in vitro [47]. Recently, lidocaine in vitro was shown to inhibit cytoskeletal remodeling and human breast cancer migration [48]. Lidocaine can also have beneficial effects on the immune system. Yardeni et al. demonstrated that a continuous infusion of lidocaine reduced plasma concentration of cytokines including IL-6, IL-8, and IL-1Ra, and improved the function of lymphocytes in response to phytohemagglutinin in patients undergoing transabdominal hysterectomy [49]. However, it is important to point out that there are no preclinical studies to show that clinically available doses can improve the oncologic outcome. Ongoing studies dealing with IV infusion of lidocaine are promising and may improve the outcome of patients undergoing cancer surgery. But some anesthetic agents may also be potentially deleterious.

### 4.2. Volatile or Inhaled Anesthesia and Cancer Cell Biology

Among several perioperative factors, some anesthetic agents may promote tumorigenesis and metastatic recurrence. Indeed, it has been shown that isoflurane can increase hypoxia inducible factor (HIF) [50], promoting angiogenesis through vascular endothelial growth factor (VEGF) signaling, likely accelerating cancer progression [51]. Molecular entities that are responsible for angiogenesis, migration, invasion, proliferation and even chemoresistance have been shown to be significantly enhanced by isoflurane through potentiation of the tumorigenic PI3K/Akt/mTOR cell signaling pathway and then in turn up-regulation of HIFs, with effects sustained for up to 24 h after a two-hour isoflurane exposure [51]. In contrast, propofol has been shown to antagonise these signaling pathways, as evidenced by a reduction in cellular levels of phosphorylated-Akt and HIF-1 alpha expression. Interestingly, propofol has also been shown to suppress the pro-malignant effects of isoflurane in vitro [51]. In line with these findings of in vitro work, retrospective clinical studies [52,53,54,55] (see below Section 4.3) also showed that TIVA is better than inhalational anesthetics during cancer surgery regarding long-term survival.

### 4.3. Retrospective and On-Going Randomized Clinical Trials on TIVA versus Inhalation

Retrospective clinical studies evaluated postoperative outcome after total intravenous anesthesia (TIVA) versus inhalation anesthesia in cancer patients. In their study including approximately 5,200 patients undergoing surgery for different types of cancer, Wigmore et al. found that inhalational (volatile) anesthesia was associated with a hazard ratio of 1.46 after multivariate analysis when compared with TIVA [52]. Recently, Zheng et al. found the same results, an association between better survival and TIVA in patients with gastric cancer undergoing surgery [53]. Lee et al. reported a better 5 years outcome after TIVA for radical mastectomy [54] and Jun et al. a better outcome in patients undergoing esophageal cancer surgery and TIVA [55]. A recent meta-analysis of these four retrospective clinical studies suggested that TIVA might be the preferred anesthetic choice in cancer surgery; however, the quality of evidences is low [56]. In contrast with these findings, a recent retrospective study in patients with lung cancer found no benefit in outcome after TIVA when compared with inhalation anesthesia [57]. To make the picture more complex, it should be pointed out that some studies have shown differences in overall survival, with no change in relapse-free survival. This may arise as a result of systemic opioid analgesics (i.e., opioid-related cardiovascular events, nocturnal arrhythmia, disordered breathing and/or infections). 

Several prospective clinical studies comparing the outcome after TIVA versus inhalation anesthesia for cancer surgery are currently under development or in progress. The effects of regional analgesia/anesthesia and of local anesthetics are also being assessed in some RCTs. 

Most prospective RCTs compare TIVA ± regional anesthesia with inhalation anesthesia for breast and colon cancer surgery (NCT2089178, NCT00938171, NCT418457, NCT2005770, NCT01975064). Other types of cancer are also being studied e.g., pancreatic cancer (NCT2335151) [58]. There are a few studies comparing the effects of TIVA versus inhalation anesthesia and of lidocaine infusion on long-term outcomes and recurrence rates for breast and colon cancer (NCT02839668, NCT02786329, NCT01204242). 

Finally, we will consider the possible role of opioids, in negative outcomes for patients undergoing cancer surgery.

### 4.4. Opioids and Cancer Recurrence

Most experimental research, consisting of both cancer cell culture studies and in vivo animal models of cancer, suggest that opioids facilitate cancer cell proliferation (such as migration, invasion, and angiogenesis) and therefore could be detrimental in cancer patients [59,60,61]. However, some laboratory studies have shown the opposite effect i.e., opioids may have an inhibitory effect on cancer cells and, depending on the animal model used, may not always enhance tumor growth. On balance the evidence does not justify reduced perioperative opioid use, which could potentially lead to worse outcomes due to the negative impact of pain [62].

There are several observational studies with conflicting results, the majority favoring an association between opioid use during cancer surgery and increased risk of recurrence [63]. Also noteworthy is the apparent association between use of the peripheral opioid antagonist methylnaltrexone, with longer survival in cancer patients who receive opioids [64]. What is needed are prospective, randomized, controlled clinical trials comparing opioid based analgesia perioperatively during tumor resection surgery of curative intent with alternative anesthetic-analgesic techniques, with long-term oncologic outcomes (recurrence free survival, overall survival) as the clinical endpoints, in combination with biological ones. Such studies take years to complete patient enrolment and follow-up, and one such trial (NCT00418457) may report the results in 2019. 

### 4.5. Opioid-Free Anesthesia and Cancer

Opioid-Free Anesthesia (OFA) based on the pre-, intra-, and postoperative administration of analgesics known to prolong disease-free survival e.g., NSAIDs and LA might be a promising therapeutic strategy for surgical cancer patients. OFA strategies for this group of patients may include the use of propofol as a hypnotic, analgesia based on neuroaxial or locoregional techniques to reduce/avoid the use of opioids, intravenous infusion of lidocaine from anesthesia induction to the immediate postoperative period, use of beta-blockers and pre- and postoperative administration of NSAID [65]. The effectiveness of other drugs such as dexmedetomidine or ketamine in improving disease-free survival, if any, is more controversial. Further studies should be conducted in the future to assess the role of these agents in treating tumor and metastatic disease [59]. However, assessing the specific role of these drugs in preventing tumor relapse is challenging. First, it requires the performance of multicenter trials involving patients with different types of cancer for which surgery is the main therapeutic option [66]. Yet, to obtain significant results, a large sample of patients is required, and long-term follow-up is necessary, as well a complex design seeing the multiple combinations of pathological processes and therapeutic interventions. A first approach may be to compare paradigms in analgesia, e.g., epidural analgesia versus the morphine use.

### 4.6. Morphine versus Epidural Analgesia in Colorectal Cancer Surgery, Lessons from an Ongoing Trial

The role of epidural analgesia (EDA) in improving long-term outcomes following colorectal cancer surgery has been discussed [67]. One study from authors of this paper found that patients operated for rectal but not colon cancer had a significantly longer survival time when regional anesthesia was combined with general anesthesia as opposed to general anesthesia alone [68]. However, it was also found in another study that epidural anesthesia as opposed to patient-controlled intravenous morphine analgesia (PCA) does not reduce the perioperative inflammatory response to colorectal surgery [69]. The precise mechanism by which EDA might improve postoperative inflammation therefore remains unclear. And the existence of a class effect regarding the influence of opioids on cancer, if any, should not be automatically accepted, as this is not clear in the literature. In a prospective, randomized, multicenter trial in patients undergoing surgery for colorectal cancer, 280 patients were randomized to receive epidural anesthesia and analgesia for perioperative pain management or intravenous PCA morphine. The primary aim of this study was to assess the time to cancer recurrence (by CT performed yearly) or all-cause mortality. Interestingly, as surgical techniques evolve, there is stratification to laparoscopic surgery/open-surgery and colon/rectal surgery. The study is expected to be complete by the end 2019 and all patients will be followed-up for a period of 5 years.

Retrospective data on patients operated for rectal cancer surgery points to favorable outcome in patients having epidural analgesia. No prospective study with an adequate number of patients has yet been completed but important findings are awaited. Another promising adjunct to analgesia is the NSAIDs, especially in the context of inflammation and COX-2 dependent mechanisms. 

### 4.7. Inflammation and NSAIDs

The story of the NSAIDs in this context began with the identification of a bimodal pattern of hazard of relapse among early stage breast cancer patients, in multiple databases. Using computer simulation and access to a high-quality database from Milan for patients treated with mastectomy only, it was proposed that relapses within 3 years of surgery are stimulated somehow by the surgical procedure. Retrospective breast cancer data suggested a plausible mechanism. Use of ketorolac, a common NSAID analgesic, was associated with superior disease-free survival [40,70]. The expected prominent early relapse events in months 9 to 18 were reduced 5-fold. Transient systemic inflammation accompanying surgery (identified by IL-6 in serum) could facilitate angiogenesis of dormant micrometastases and proliferation of dormant single cells and could have been effectively blocked by the perioperative NSAID. If this observation holds up to further scrutiny, it could mean that the use of this safe, inexpensive and effective anti-inflammatory agent at surgery might substantially reduce early relapses and reduce mortality by 25–50%. 

The findings have recently been confirmed in a mouse-model by Krall et al. [71] and in a retrospective study by Desmedt et al. [72]. 

However, the Ketorolac in Breast Cancer trial (NCT01806259), a multicenter, placebo-controlled, randomized phase III trial in high-risk breast cancer patients, has recently shown that a single dose of ketorolac tromethamine versus placebo 30 min before surgery, does not increase disease-free survival in high risk breast cancer patients [73,74]. Overall survival is also comparable. As no safety concerns were observed, multiple doses could be studied during the perioperative period. Considering these inconsistent results, it can be suggested that this intervention should be studied further in future randomized controlled clinical trials.

## 5. Other Perioperative Interventions

### 5.1. Perioperative Nutrition

Perioperative interventions that may have a positive effect on the outcome also include other factors than anesthetic/analgesic techniques. Proper, patient-tailored nutrition plays a major role in all kinds of cancer surgery. Its beneficial effect on short- and long-term outcomes is multifactorial. Not only does it help the patient to recover faster from surgical trauma, but also decreases the number and severity of postoperative complications [75]. Non-complicated postoperative outcomes enable timely administration of oncological treatment as well as reduce the risk of applying suboptimal radio- and chemotherapy doses, which happen in nutritionally compromised patients [76]. It has also been shown that patients who suffer from postoperative complications have worse overall disease-free survival [77]. 

Well-planned preoperative nutrition also helps to prepare a malnourished patient for surgery and oncological treatment. Whenever possible, enteral or oral route is preferred as it helps to maintain gastrointestinal physiology and mucosal integrity. For patients in whom enteral route is not adequate, supplemental parenteral nutrition may be used. Patients without mechanical obstruction of gastrointestinal tract should be prescribed oral nutritional supplements (ONS) as standard part of their preoperative nutritional preparation. Standard ONS use has been shown to be effective in both malnourished and non-malnourished, weight losing patients [78]. 

The Enhanced Recovery After Surgery (ERAS) paradigm has been proven safe and beneficial in most surgical fields, including surgical oncology. The paradigm should be applied in all surgical cancer patients, with early oral feeding where applicable and fluid restriction therapy [79,80].

Briefly, a modern, patient-tailored nutritional approach aimed at reducing nutritional deficiencies and minimization of the fasting period should be a part of oncological treatment plan. 

### 5.2. Perioperative Nutrition Protocol

The nutritional status of the patient significantly influences the outcome of the treatment, whether it is obesity or malnutrition with loss of muscle mass [81]. Inadequate dietary support in the perioperative period compromises the surgical outcome [82]. Up to 30 percent of patients hospitalized for surgical oncology with major surgery need nutritional support according to NSR 2002 evaluation [83]. Bioimpedance measurements are adding quality and quantity data to nutritional status gained through questionnaires and body mass index (BMI) data [84].

Some experiences show that it is possible to evaluate the patients multiple times by a nurse dealing with nutrition in consultation with the anesthesiologist. For example, the first appointment may be directly after the surgery counselling when the surgeon sets the date of surgery (usually two weeks before the operation), on the day of surgery, upon leaving the intensive care unit (3rd day), on discharge from the hospital and the 15th and 30th postoperative day. The nurse can take bioimpedance measurements in addition to patient history on nutritional habits and NSR 2002. Preliminary results show that the data gained through NSR 2002, bioimpedance and primary laboratory results can be effectively used to tailor individual nutrition plan either enteral or parenteral. Bioimpedance proved to be a useful additional surveillance tool as it added objectiveness to muscle loss evaluation which is often an underestimated marker of malnutrition, even if loss of muscle mass due to cancer cachexia would not be easily differentiated [85]. This can be done conjointly to a preoperative physical rehabilitation, called prehabilitation.

### 5.3. Prehabilitation

The individual’s level of physical fitness prior to an operation may be predictive of postoperative outcome: physical fitness is an overall term encompassing aerobic capacity, endurance, muscular strength, and body composition [86,87,88,89,90]. Prehabilitation i.e., improving the functional capacity of an individual to withstand a stressful event, may improve postoperative outcome and decrease the associated hospital costs of operative intervention while enhancing patient quality of life [91,92]. A recent review has shown that preoperative exercise therapy may improve physical fitness of patients prior to a major abdominal operation although the ability to improve postoperative outcomes is unclear due to the lack of randomized controlled trials [90].

A recent meta-analysis by Moran et al. has shown that prehabilitation, consisting of inspiratory muscle training, aerobic exercise, and resistance training, appears to decrease the incidence of all postoperative complications in patients undergoing intra-abdominal operations (Odds Ratio, OR: 0.59, 95% CI: 0.38–0.91) [93]. This effect was strongest when prehabilitation was compared with usual care or breathing exercises only (OR: 0.35, 95% CI: 0.17–0.71). Furthermore, prehabilitation significantly decreased the incidence of postoperative pulmonary complications (OR: 0.27, 95% CI: 0.13–0.57), which were measured as the primary complication of interest in the majority of studies reviewed. 

The main obstacle to the spreading of nutritional therapy as of prehabilitation is compliance. Current standards in cancer care demand minimal delay between cancer diagnosis and treatment, thus incorporating these interventions into a time-restricted clinical pathway may be challenging.

To overcome this limit, prehabilitation and ERAS, both aimed to minimize operative stress and complications, can be integrated. However, the literature specific to ERAS and prehabilitation is limited, and the two programs are seldom combined [94]. The designing of further RCTs should be considered essential in this field. The important question remains, does prehabilitation improve outcomes?

### 5.4. Does Prehabilitation Reduce the Risk for Cancer Recurrence and/or Improves Survival?

The aging population has an increased prevalence of cancers in the setting of a progressive state of vulnerability (frailty). Homeostenosis is the narrowing of the homeostatic functional capacity of an organism. Deconditioning is the decreased ability to adapt to normal conditions, which can also occur in patients with cancer. Therefore, under these conditions, morbidity and mortality after cancer surgery is progressively increased and impaired because of the cancer itself. The individual’s level of physical fitness prior to an operation may be predictive of postoperative outcome: physical fitness is an overall term encompassing aerobic capacity, endurance, muscular strength and body composition [86].

Physical activity is known to reduce the risk for cancer development. Moreover, physical activity performed both before surgery (prehabilitation), or after cancer surgery (rehabilitation), improves immune function and metabolism, reduces length of stay, disability, complications, early retirement, all-cause mortality, and even recurrence and mortality from some cancer types [95,96,97]. In the above-mentioned meta-analysis by Moran et al., the authors showed that inspiratory muscle training, aerobic exercise, and resistance training, appears to decrease the incidence of all postoperative complications, but does not influence postoperative LOS [93]. However, the number of available studies was small and the methodological quality can be debated. For these reasons, prehabilitation and rehabilitation (inspiratory muscle training, aerobic exercise, resistance training) remain promising strategies that deserve to be better studied in randomized trials and in association with other strategies for reducing recurrence and increasing survival after cancer surgery, probably focusing on the right patients subgroups, that need to be identified. 

## 6. Conclusions

In conclusion, the hypothesis that anesthesia, analgesia and other perioperative interventions such as optimizing nutrition and prehabilitation might influence cancer biology and therefore patient outcomes is supported by some preclinical and clinical studies. The conflicting results highlight the need to establish a large consortium, using innovative approaches, while linking data from multiple sources and multiple techniques, and to undertake prospective, randomized, multicenter clinical trials.

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
