# Peer review of "How Anesthetic, Analgesic and Other Non-Surgical Techniques During Cancer Surgery Might Affect Postoperative Oncologic Outcomes: A Summary of Current State of Evidence"

_cancers, 2019, doi:10.3390/cancers11050592_

Round 1
Reviewer 1 Report
The paper has multiple authors from multiple centres, and attempts to provide an overview of the evidence of analgesic/anaesthesia in cancer surgery. However, the topic is large and broad, making it very difficult to summarise so much information in this relatively short paper.
Many references are used, but it was unclear how they were selected. Even though the abstract admits that this was a ‘narrative review’ the title implies a larger assimilation of the evidence that is more systematic than that provided: ‘summary of the current state of evidence’. A common issue with narrative reviews is that the choice of references and hence evidence is often greatly influenced by the authors’ own knowledge, preferences and awareness of individual studies.
For example, in the section on opioids and recurrence, there is a statement that opioid use might be associated with increased risk of recurrence, but this is linked to a single reference 65 which is a small study of lung cancer tissue samples. It appears that the statement comes from the Background section of reference 65, which did not make any conclusion about an association because studies were either positive or negative.
There is a section on RCTs of TIVA vs inhalation with reference to survival. But there are systematic reviews on anaesthesia techniques and other outcomes such as post-operative pain, which seem relevant to patient outcomes but not mentioned. The section could also have covered other aspects of TIVA vs inhalation, because there are reasons for using either method.
There is a sizeable section on peri-operative interventions that is quite different to analgesic/ anaesthesia techniques (and therefore the title), so its relevance in the paper is unclear; it is a topic in its own right.
Author Response
Response to Reviewer 1 Comments
Point 1: The paper has multiple authors from multiple centres, and attempts to provide an overview of the evidence of analgesic/anaesthesia in cancer surgery. However, the topic is large and broad, making it very difficult to summarise so much information in this relatively short paper.
Response 1: We agree with the comments. As detailed in other responses to this and reviewer 2 and 3, we have tried to clarify the given information.
Point 2: Many references are used, but it was unclear how they were selected. Even though the abstract admits that this was a ‘narrative review’ the title implies a larger assimilation of the evidence that is more systematic than that provided: ‘summary of the current state of evidence’. A common issue with narrative reviews is that the choice of references and hence evidence is often greatly influenced by the authors’ own knowledge, preferences and awareness of individual studies.
Response 2: This is indeed a narrative review. The way used to deal with this type of issue is now clarified at the end of the introduction section: “After a two-rounds consultation of all the members, aiming to have a complete vision, updates were proposed and consensually accepted by the entire panel. Each topic is illustrated by highest level of evidence works.”
Point 3: For example, in the section on opioids and recurrence, there is a statement that opioid use might be associated with increased risk of recurrence, but this is linked to a single reference 65 which is a small study of lung cancer tissue samples. It appears that the statement comes from the Background section of reference 65, which did not make any conclusion about an association because studies were either positive or negative.
Response 3: This reviewer is right, as the statement is inadequately referenced. We have moved to a better reference: Kim R. Effects of surgery and anesthetic choice on immunosuppression and cancer recurrence. J Transl Med. 2018 Jan 18;16(1):8.
Point 4: There is a section on RCTs of TIVA vs inhalation with reference to survival. But there are systematic reviews on anaesthesia techniques and other outcomes such as post-operative pain, which seem relevant to patient outcomes but not mentioned. The section could also have covered other aspects of TIVA vs inhalation, because there are reasons for using either method.
Response 4: We agree with the reviewer but have tried to be as clear as possible. To not confuse the reader, we propose to not complexify the paragraph.
Point 5: There is a sizeable section on peri-operative interventions that is quite different to analgesic/ anaesthesia techniques (and therefore the title), so its relevance in the paper is unclear; it is a topic in its own right.
Response 5: Indeed, the section, seemed as important by the panel, is not well presented in the title. In the title, the abstract and the introduction sections, we have now clearly mentioned the anaesthetic, analgesic and other non-surgical techniques/interventions. We would be happy to consider any other suggestion.
Reviewer 2 Report
This is well-written manuscript and give lots of updated information about anesthesia and oncologic outcomes, which is emerging part in the field of anesthesiology. But there are some minor comments regarding manuscript.
References 12 is inadequate, unlike the text, it is related to opioids.
References 13 is a paper comparing the effects of TIVA with balanced anesthesia on VEGF, and so on. it is also inadequate to the text.
Metastasis: a role for amide-type local anaesthetics?
Although the perioperative use of lidocaine is promising method for the oncologic outcome as described by the authors, there are no preclinical studies to show that clinically available doses can improve the oncologic outcome. Considering systemic toxicity of lidocaine, it would be better to clarify this point.
Retrospective and on-going randomized clinical trials on TIVA vs. inhalation
Even there have been many reports to show TIVA is superior than inhalation in oncologic outcome, recent two retrospective studies in the patients with breast cancer showed that there were no association between type of anesthesia used and the long-term prognosis of breast cancer. (Anesthesiology. 2019 Jan;130(1):31-40, Oncotarget. 2017 Sep 18;8(52):90477-90487.) I think it would be better to mention these two study. I think that inconsistent results may occur depending on the cancer cell type. How do the authors think?
Opioids and cancer recurrence
The possibility of an increased risk of cancer recurrence due to the oversecretion of steroids or catecholamines as a stress response due to pain is something that surgeons, anesthesiologists and patients are all deeply concerned about. As reducing the use of opioids can have a negative impact on the oncologic outcome by aggravating the pain, we have to consider both aspects, which can lead to the opposite result. It would be better to add the authors' view on pain and opioid in the paragraph discussing the use of opioids in the perioperative period and oncologic outcomes.
Author Response
Response to Reviewer 2 Comments
Point 1: This is well-written manuscript and give lots of updated information about anesthesia and oncologic outcomes, which is emerging part in the field of anesthesiology. But there are some minor comments regarding manuscript.
Response 1: We thank the reviewer for the encouraging comment.
Point 2: References 12 is inadequate, unlike the text, it is related to opioids.
References 13 is a paper comparing the effects of TIVA with balanced anesthesia on VEGF, and so on. it is also inadequate to the text.
Response 2: The reviewer is right. The references 12 and 13 were accidentally replaced by others. This is now corrected.
Point 3: Metastasis: a role for amide-type local anaesthetics?
Although the perioperative use of lidocaine is promising method for the oncologic outcome as described by the authors, there are no preclinical studies to show that clinically available doses can improve the oncologic outcome. Considering systemic toxicity of lidocaine, it would be better to clarify this point.
Response 3: We are sorry to not have been sufficiently clear. We have added a sentence to clarify as suggested: However, it is important to point that there are no preclinical studies to show that clinically available doses can improve the oncologic outcome.
Point 4: Retrospective and on-going randomized clinical trials on TIVA vs. inhalation
Even there have been many reports to show TIVA is superior than inhalation in oncologic outcome, recent two retrospective studies in the patients with breast cancer showed that there were no association between type of anesthesia used and the long-term prognosis of breast cancer. (Anesthesiology. 2019 Jan;130(1):31-40, Oncotarget. 2017 Sep 18;8(52):90477-90487.) I think it would be better to mention these two study. I think that inconsistent results may occur depending on the cancer cell type. How do the authors think?
Response 4: We totally that there may be a tumour-specific effect, but also that there are many potential other confounding factors. To clarify this issue, we have mentioned some of the inconsistencies seen in the literature, and hope that this may be in line with the reviewer point of view. Here, we have tried to put the emphasis more on the ongoing RCTs, as the retrospective studies are more hypothesis-generating.
Point 5: Opioids and cancer recurrence
The possibility of an increased risk of cancer recurrence due to the oversecretion of steroids or catecholamines as a stress response due to pain is something that surgeons, anesthesiologists and patients are all deeply concerned about. As reducing the use of opioids can have a negative impact on the oncologic outcome by aggravating the pain, we have to consider both aspects, which can lead to the opposite result. It would be better to add the authors' view on pain and opioid in the paragraph discussing the use of opioids in the perioperative period and oncologic outcomes.
Response 5: Even if not very clear were this paragraph may be better placed, we totally agree that pain treatment is a priority, and even a best practice. We have, accordingly to the reviewer’s suggestion, added a specific sentence for: This should lead to not reduce too far the use of opioids, potentially leading to the opposite result as aggravating the pain can have a negative impact on the outcome.
Reviewer 3 Report
Critique
This is an important review. There are some syntax problems.
On page 4, line 150, ‘neurons by pain to enhanced tumor progression” should be “activated sensory neurons induced by pain in hands tumor progression”.
On page 4, lying 171 “effecting” should be “inducing, and tumor angiogenesis”.
On page for line 173 it should be” Resolvin D1, a metabolite of Omega -3 fatty acids through the Cox -2 pathway…”
On page for line 175 it should be “perioperative treatment with Resolvin D1 in a murine a model reduces lung metastases”
On page for line 183 “They” technically would refer back to Yuma in research. However the authors intend “they” to refer to animal models.
On page 5 line 204 it should be “Lidocaine at clinically achievable concentrations…”
In the section retrospective an ongoing randomized trial, the authors should distinguish between cancer specific survival or and overall survival. Several studies have demonstrated no change in relapse free survival but reduced overall survival. This may arises as a direct result of systemic opioid analgesics. Opioids may induce cardiovascular events, sleep disordered breathing with nocturnal arrhythmias, systemic infections and delayed wound healing. Even tramadol has been associated with increased mortality1,2. The mortality may be accentuated by comorbidities and the deaths not overdose deaths3-5. The deaths may be infectious and not be recognized as a potential adverse effect of opioid analgesics.
On page 6 in the section in titled opioids and cancer recurrence. There is a body of evidence that met-in Kaplan also known as opioid growth factor binds to opioid growth factor receptors found in cytoplasm and has anti a cancer activity. The real question is whether the effect noted by opioid alkaloids is a class effect? For instance,6-20 buprenorphine is a “partial agonist” at the immune receptor and lax immediate no suppression. Morphine is a full act and is and is immunosuppressive as is Fentanyl. Is the risk of recurrence the same with non-immunosuppressive opioids as it is with immunosuppressive opioids?
On page 7 line 293 “it should be “by which EDA might improve” ( singular)
In the section in titled morphine verses epidural analgesia again the authors are assuming a drug class effect which is likely not to be the case. Morphine is in immunosuppressive drug, hydromorphone, oxycodone and buprenorphine are not.
In the section in titled perioperative nutrition, it is apparent that the gut micro bio moist is a role in inflammation and immunity as well as other disease entities. Nutrition will influence the micro by on. Morphine causes loss of got barrier function with an increase risk of infections or inflammation.21-26
Bioelectrical impedance reflex the phase angle measure. The phase angle is a combination of muscle mass and cellular membrane integrity. Loss of muscle mass may be from cancer cachexia rather than poor nutrition. The bioimpedance will not be able to decipher the difference. The Patient-Generated Subjective Global Assessment 27-30and the Glasgow Prognostic Scale may be helpful in this regard31-37
1. Zeng C, Dubreuil M, LaRochelle MR, et al. Association of Tramadol With All-Cause Mortality Among Patients With Osteoarthritis. Jama. 2019;321(10):969-982.
2. Ray WA, Chung CP, Murray KT, Hall K, Stein CM. Prescription of Long-Acting Opioids and Mortality in Patients With Chronic Noncancer Pain. Jama. 2016;315(22):2415-2423.
3. Vozoris NT, Wang X, Austin PC, et al. Adverse cardiac events associated with incident opioid drug use among older adults with COPD. European journal of clinical pharmacology. 2017;73(10):1287-1295.
4. Vozoris NT, Wang X, Fischer HD, et al. Incident opioid drug use and adverse respiratory outcomes among older adults with COPD. The European respiratory journal. 2016;48(3):683-693.
5. Wiese AD, Grijalva CG. The use of prescribed opioid analgesics & the risk of serious infections. Future microbiology. 2018;13:849-852.
6. Sacerdote P. Opioids and the immune system. Palliat Med. 2006;20 Suppl 1:s9-15.
7. Vallejo R, de Leon-Casasola O, Benyamin R. Opioid therapy and immunosuppression: a review. Am J Ther. 2004;11(5):354-365.
8. Martucci C, Panerai AE, Sacerdote P. Chronic fentanyl or buprenorphine infusion in the mouse: similar analgesic profile but different effects on immune responses. Pain. 2004;110(1-2):385-392.
9. Ernst G, Pfaffenzeller P. [Effect on morphine and other opioids on immune function]. Schmerz. 1998;12(3):187-194.
10. Wei G, Moss J, Yuan CS. Opioid-induced immunosuppression: is it centrally mediated or peripherally mediated? Biochemical pharmacology. 2003;65(11):1761-1766.
11. Pruett SB, Han YC, Fuchs BA. Morphine suppresses primary humoral immune responses by a predominantly indirect mechanism. The Journal of pharmacology and experimental therapeutics. 1992;262(3):923-928.
12. Kim JY, Ahn HJ, Kim JK, Kim J, Lee SH, Chae HB. Morphine Suppresses Lung Cancer Cell Proliferation Through the Interaction with Opioid Growth Factor Receptor: An In Vitro and Human Lung Tissue Study. Anesthesia and analgesia. 2016;123(6):1429-1436.
13. Kren NP, Zagon IS, McLaughlin PJ. Mutations in the opioid growth factor receptor in human cancers alter receptor function. International journal of molecular medicine. 2015;36(1):289-293.
14. Zagon IS, McLaughlin PJ. Opioid growth factor and the treatment of human pancreatic cancer: a review. World journal of gastroenterology. 2014;20(9):2218-2223.
15. Zagon IS, Porterfield NK, McLaughlin PJ. Opioid growth factor - opioid growth factor receptor axis inhibits proliferation of triple negative breast cancer. Experimental biology and medicine. 2013;238(6):589-599.
16. McLaughlin PJ, Zagon IS. The opioid growth factor-opioid growth factor receptor axis: homeostatic regulator of cell proliferation and its implications for health and disease. Biochemical pharmacology. 2012;84(6):746-755.
17. Fanning J, Hossler CA, Kesterson JP, Donahue RN, McLaughlin PJ, Zagon IS. Expression of the opioid growth factor-opioid growth factor receptor axis in human ovarian cancer. Gynecologic oncology. 2012;124(2):319-324.
18. Donahue RN, McLaughlin PJ, Zagon IS. Under-expression of the opioid growth factor receptor promotes progression of human ovarian cancer. Experimental biology and medicine. 2012;237(2):167-177.
19. Donahue RN, McLaughlin PJ, Zagon IS. The opioid growth factor (OGF) and low dose naltrexone (LDN) suppress human ovarian cancer progression in mice. Gynecologic oncology. 2011;122(2):382-388.
20. Smith JP, Bingaman SI, Mauger DT, Harvey HH, Demers LM, Zagon IS. Opioid growth factor improves clinical benefit and survival in patients with advanced pancreatic cancer. Open Access J Clin Trials. 2010;2010(2):37-48.
21. Lee K, Vuong HE, Nusbaum DJ, Hsiao EY, Evans CJ, Taylor AMW. The gut microbiota mediates reward and sensory responses associated with regimen-selective morphine dependence. Neuropsychopharmacology : official publication of the American College of Neuropsychopharmacology. 2018;43(13):2606-2614.
22. Sindberg GM, Callen SE, Banerjee S, et al. Morphine Potentiates Dysbiotic Microbial and Metabolic Shifts in Acute SIV Infection. Journal of neuroimmune pharmacology : the official journal of the Society on NeuroImmune Pharmacology. 2018.
23. Wang F, Meng J, Zhang L, Johnson T, Chen C, Roy S. Morphine induces changes in the gut microbiome and metabolome in a morphine dependence model. Scientific reports. 2018;8(1):3596.
24. Kang M, Mischel RA, Bhave S, et al. The effect of gut microbiome on tolerance to morphine mediated antinociception in mice. Scientific reports. 2017;7:42658.
25. Banerjee S, Sindberg G, Wang F, et al. Opioid-induced gut microbial disruption and bile dysregulation leads to gut barrier compromise and sustained systemic inflammation. Mucosal immunology. 2016;9(6):1418-1428.
26. Meng J, Banerjee S, Li D, et al. Opioid Exacerbation of Gram-positive sepsis, induced by Gut Microbial Modulation, is Rescued by IL-17A Neutralization. Scientific reports. 2015;5:10918.
27. Na BG, Han SS, Cho YA, et al. Nutritional Status of Patients with Cancer: A Prospective Cohort Study of 1,588 Hospitalized Patients. Nutrition and cancer. 2019:1-9.
28. Song C, Cao J, Zhang F, et al. Nutritional Risk Assessment by Scored Patient-Generated Subjective Global Assessment Associated with Demographic Characteristics in 23,904 Common Malignant Tumors Patients. Nutrition and cancer. 2019:1-11.
29. Koshimoto S, Arimoto M, Saitou K, et al. Need and demand for nutritional counselling and their association with quality of life, nutritional status and eating-related distress among patients with cancer receiving outpatient chemotherapy: a cross-sectional study. Support Care Cancer. 2019.
30. Wu M, Lian XJ, Jia JM, et al. The role of the Patient-Generated Subjective Global Assessment (PG-SGA) and biochemical markers in predicting anemia patients with cancer. Support Care Cancer. 2019;27(4):1443-1448.
31. Imrie CW. Host systemic inflammatory response influences outcome in pancreatic cancer. Pancreatology : official journal of the International Association of Pancreatology. 2015;15(4):327-330.
32. Jiang AG, Chen HL, Lu HY. The relationship between Glasgow Prognostic Score and serum tumor markers in patients with advanced non-small cell lung cancer. BMC cancer. 2015;15:386.
33. Dreanic J, Maillet M, Dhooge M, et al. Prognostic value of the Glasgow Prognostic Score in metastatic colorectal cancer in the era of anti-EGFR therapies. Med Oncol. 2013;30(3):656.
34. Stotz M, Gerger A, Eisner F, et al. Increased neutrophil-lymphocyte ratio is a poor prognostic factor in patients with primary operable and inoperable pancreatic cancer. British journal of cancer. 2013;109(2):416-421.
35. Jeong JH, Lim SM, Yun JY, et al. Comparison of two inflammation-based prognostic scores in patients with unresectable advanced gastric cancer. Oncology. 2012;83(5):292-299.
36. Fujiwara Y, Shiba H, Furukawa K, et al. Glasgow prognostic score is related to blood transfusion requirements and post-operative complications in hepatic resection for hepatocellular carcinoma. Anticancer research. 2010;30(12):5129-5136.
37. Al Murri AM, Bartlett JM, Canney PA, Doughty JC, Wilson C, McMillan DC. Evaluation of an inflammation-based prognostic score (GPS) in patients with metastatic breast cancer. British journal of cancer. 2006;94(2):227-230.
Author Response
Response to Reviewer 3 Comments
Point 1: This is an important review. There are some syntax problems.
Response 1: We thank the reviewer for the encouraging comment and have followed all the suggestions regarding the syntax problems. Additionally, we have entirely revised the text and rewritten some paragraphes/sentences.
Point 2: On page 4, line 150, ‘neurons by pain to enhanced tumor progression” should be “activated sensory neurons induced by pain in hands tumor progression”.
On page 4, lying 171 “effecting” should be “inducing, and tumor angiogenesis”.
On page for line 173 it should be” Resolvin D1, a metabolite of Omega -3 fatty acids through the Cox -2 pathway…”
On page for line 175 it should be “perioperative treatment with Resolvin D1 in a murine a model reduces lung metastases”
On page for line 183 “They” technically would refer back to Yuma in research. However the authors intend “they” to refer to animal models.
On page 5 line 204 it should be “Lidocaine at clinically achievable concentrations…”
Response 2: We thank again the reviewer and have modified all the sentences.
Point 3: In the section retrospective an ongoing randomized trial, the authors should distinguish between cancer specific survival or and overall survival. Several studies have demonstrated no change in relapse free survival but reduced overall survival. This may arises as a direct result of systemic opioid analgesics. Opioids may induce cardiovascular events, sleep disordered breathing with nocturnal arrhythmias, systemic infections and delayed wound healing. Even tramadol has been associated with increased mortality1,2. The mortality may be accentuated by comorbidities and the deaths not overdose deaths3-5. The deaths may be infectious and not be recognized as a potential adverse effect of opioid analgesics.
Response 2: This is a very valuable comments. We have inserted a sentence to point this and are impressed by the referencing of the reviewer. We are sorry for not having the place to introduce all the reviewer’s references that may take place for an entire article.
Point 3: On page 6 in the section in titled opioids and cancer recurrence. There is a body of evidence that met-in Kaplan also known as opioid growth factor binds to opioid growth factor receptors found in cytoplasm and has anti a cancer activity. The real question is whether the effect noted by opioid alkaloids is a class effect? For instance,6-20 buprenorphine is a “partial agonist” at the immune receptor and lax immediate no suppression. Morphine is a full act and is and is immunosuppressive as is Fentanyl. Is the risk of recurrence the same with non-immunosuppressive opioids as it is with immunosuppressive opioids?
Response 3: The question is of prime importance. However, the literature does not permit to give a clear answer, especially regarding the place we have in this broad review.
Point 4: On page 7 line 293 “it should be “by which EDA might improve” ( singular)
Response 4: Corrected.
Point 5: In the section in titled morphine verses epidural analgesia again the authors are assuming a drug class effect which is likely not to be the case. Morphine is in immunosuppressive drug, hydromorphone, oxycodone and buprenorphine are not.
Response 5: We agree with the reviewer that the existence of a class effect regarding the influence of opioids on cancer, if any, should not be automatically accepted, as not clear in the literature. This is now added in the paragraph.
Point 6: In the section in titled perioperative nutrition, it is apparent that the gut micro bio moist is a role in inflammation and immunity as well as other disease entities. Nutrition will influence the micro by on. Morphine causes loss of got barrier function with an increase risk of infections or inflammation.21-26
Response 6: This is a very interesting comment, but, at this stage, difficult to introduce in this first introductory paper for many readers. We propose to let it as a ‘second step’, i.e. not in this paper.
Point 7: Bioelectrical impedance reflex the phase angle measure. The phase angle is a combination of muscle mass and cellular membrane integrity. Loss of muscle mass may be from cancer cachexia rather than poor nutrition. The bioimpedance will not be able to decipher the difference. The Patient-Generated Subjective Global Assessment 27-30and the Glasgow Prognostic Scale may be helpful in this regard31-37
Response 6: We totally agree with the reviewer comment regarding the limits of the bioimpedance and have introduced a sentence to point this. In fact, this paragraph just points an illustrative experience.
1. Zeng C, Dubreuil M, LaRochelle MR, et al. Association of Tramadol With All-Cause Mortality Among Patients With Osteoarthritis. Jama. 2019;321(10):969-982.
2. Ray WA, Chung CP, Murray KT, Hall K, Stein CM. Prescription of Long-Acting Opioids and Mortality in Patients With Chronic Noncancer Pain. Jama. 2016;315(22):2415-2423.
3. Vozoris NT, Wang X, Austin PC, et al. Adverse cardiac events associated with incident opioid drug use among older adults with COPD. European journal of clinical pharmacology. 2017;73(10):1287-1295.
4. Vozoris NT, Wang X, Fischer HD, et al. Incident opioid drug use and adverse respiratory outcomes among older adults with COPD. The European respiratory journal. 2016;48(3):683-693.
5. Wiese AD, Grijalva CG. The use of prescribed opioid analgesics & the risk of serious infections. Future microbiology. 2018;13:849-852.
6. Sacerdote P. Opioids and the immune system. Palliat Med. 2006;20 Suppl 1:s9-15.
7. Vallejo R, de Leon-Casasola O, Benyamin R. Opioid therapy and immunosuppression: a review. Am J Ther. 2004;11(5):354-365.
8. Martucci C, Panerai AE, Sacerdote P. Chronic fentanyl or buprenorphine infusion in the mouse: similar analgesic profile but different effects on immune responses. Pain. 2004;110(1-2):385-392.
9. Ernst G, Pfaffenzeller P. [Effect on morphine and other opioids on immune function]. Schmerz. 1998;12(3):187-194.
10. Wei G, Moss J, Yuan CS. Opioid-induced immunosuppression: is it centrally mediated or peripherally mediated? Biochemical pharmacology. 2003;65(11):1761-1766.
11. Pruett SB, Han YC, Fuchs BA. Morphine suppresses primary humoral immune responses by a predominantly indirect mechanism. The Journal of pharmacology and experimental therapeutics. 1992;262(3):923-928.
12. Kim JY, Ahn HJ, Kim JK, Kim J, Lee SH, Chae HB. Morphine Suppresses Lung Cancer Cell Proliferation Through the Interaction with Opioid Growth Factor Receptor: An In Vitro and Human Lung Tissue Study. Anesthesia and analgesia. 2016;123(6):1429-1436.
13. Kren NP, Zagon IS, McLaughlin PJ. Mutations in the opioid growth factor receptor in human cancers alter receptor function. International journal of molecular medicine. 2015;36(1):289-293.
14. Zagon IS, McLaughlin PJ. Opioid growth factor and the treatment of human pancreatic cancer: a review. World journal of gastroenterology. 2014;20(9):2218-2223.
15. Zagon IS, Porterfield NK, McLaughlin PJ. Opioid growth factor - opioid growth factor receptor axis inhibits proliferation of triple negative breast cancer. Experimental biology and medicine. 2013;238(6):589-599.
16. McLaughlin PJ, Zagon IS. The opioid growth factor-opioid growth factor receptor axis: homeostatic regulator of cell proliferation and its implications for health and disease. Biochemical pharmacology. 2012;84(6):746-755.
17. Fanning J, Hossler CA, Kesterson JP, Donahue RN, McLaughlin PJ, Zagon IS. Expression of the opioid growth factor-opioid growth factor receptor axis in human ovarian cancer. Gynecologic oncology. 2012;124(2):319-324.
18. Donahue RN, McLaughlin PJ, Zagon IS. Under-expression of the opioid growth factor receptor promotes progression of human ovarian cancer. Experimental biology and medicine. 2012;237(2):167-177.
19. Donahue RN, McLaughlin PJ, Zagon IS. The opioid growth factor (OGF) and low dose naltrexone (LDN) suppress human ovarian cancer progression in mice. Gynecologic oncology. 2011;122(2):382-388.
20. Smith JP, Bingaman SI, Mauger DT, Harvey HH, Demers LM, Zagon IS. Opioid growth factor improves clinical benefit and survival in patients with advanced pancreatic cancer. Open Access J Clin Trials. 2010;2010(2):37-48.
21. Lee K, Vuong HE, Nusbaum DJ, Hsiao EY, Evans CJ, Taylor AMW. The gut microbiota mediates reward and sensory responses associated with regimen-selective morphine dependence. Neuropsychopharmacology : official publication of the American College of Neuropsychopharmacology. 2018;43(13):2606-2614.
22. Sindberg GM, Callen SE, Banerjee S, et al. Morphine Potentiates Dysbiotic Microbial and Metabolic Shifts in Acute SIV Infection. Journal of neuroimmune pharmacology : the official journal of the Society on NeuroImmune Pharmacology. 2018.
23. Wang F, Meng J, Zhang L, Johnson T, Chen C, Roy S. Morphine induces changes in the gut microbiome and metabolome in a morphine dependence model. Scientific reports. 2018;8(1):3596.
24. Kang M, Mischel RA, Bhave S, et al. The effect of gut microbiome on tolerance to morphine mediated antinociception in mice. Scientific reports. 2017;7:42658.
25. Banerjee S, Sindberg G, Wang F, et al. Opioid-induced gut microbial disruption and bile dysregulation leads to gut barrier compromise and sustained systemic inflammation. Mucosal immunology. 2016;9(6):1418-1428.
26. Meng J, Banerjee S, Li D, et al. Opioid Exacerbation of Gram-positive sepsis, induced by Gut Microbial Modulation, is Rescued by IL-17A Neutralization. Scientific reports. 2015;5:10918.
27. Na BG, Han SS, Cho YA, et al. Nutritional Status of Patients with Cancer: A Prospective Cohort Study of 1,588 Hospitalized Patients. Nutrition and cancer. 2019:1-9.
28. Song C, Cao J, Zhang F, et al. Nutritional Risk Assessment by Scored Patient-Generated Subjective Global Assessment Associated with Demographic Characteristics in 23,904 Common Malignant Tumors Patients. Nutrition and cancer. 2019:1-11.
29. Koshimoto S, Arimoto M, Saitou K, et al. Need and demand for nutritional counselling and their association with quality of life, nutritional status and eating-related distress among patients with cancer receiving outpatient chemotherapy: a cross-sectional study. Support Care Cancer. 2019.
30. Wu M, Lian XJ, Jia JM, et al. The role of the Patient-Generated Subjective Global Assessment (PG-SGA) and biochemical markers in predicting anemia patients with cancer. Support Care Cancer. 2019;27(4):1443-1448.
31. Imrie CW. Host systemic inflammatory response influences outcome in pancreatic cancer. Pancreatology : official journal of the International Association of Pancreatology. 2015;15(4):327-330.
32. Jiang AG, Chen HL, Lu HY. The relationship between Glasgow Prognostic Score and serum tumor markers in patients with advanced non-small cell lung cancer. BMC cancer. 2015;15:386.
33. Dreanic J, Maillet M, Dhooge M, et al. Prognostic value of the Glasgow Prognostic Score in metastatic colorectal cancer in the era of anti-EGFR therapies. Med Oncol. 2013;30(3):656.
34. Stotz M, Gerger A, Eisner F, et al. Increased neutrophil-lymphocyte ratio is a poor prognostic factor in patients with primary operable and inoperable pancreatic cancer. British journal of cancer. 2013;109(2):416-421.
35. Jeong JH, Lim SM, Yun JY, et al. Comparison of two inflammation-based prognostic scores in patients with unresectable advanced gastric cancer. Oncology. 2012;83(5):292-299.
36. Fujiwara Y, Shiba H, Furukawa K, et al. Glasgow prognostic score is related to blood transfusion requirements and post-operative complications in hepatic resection for hepatocellular carcinoma. Anticancer research. 2010;30(12):5129-5136.
37. Al Murri AM, Bartlett JM, Canney PA, Doughty JC, Wilson C, McMillan DC. Evaluation of an inflammation-based prognostic score (GPS) in patients with metastatic breast cancer. British journal of cancer. 2006;94(2):227-230.
Round 2
Reviewer 1 Report
None